# Induction of zinc conjugated with Doxorubicin for the prevention of aggregating β-catenin in the Wnt signaling pathway investigated through computational approaches

Gomathi Rajagopal[1‡], Balajee Ramachandran[2,3‡], Paradesi Deivanayagam[1*], Venkatesan Srinivasadesikan[4], Boomi Pandi[2], Saravanan Muthupandian[5,6], Rajamanikandan Sundarraj[7,8], Goyitom Gebremedhn Gebru[9*]

1 Department of Chemistry, SRM Institute of Science and Technology, Chennai, Tamil Nadu, India, 2 Structural Biology and Biocomputing Lab, Department of Bioinformatics, Alagappa University, Karaikudi, Tamil Nadu, India, 3 Department of Pharmacology, Physiology & Biophysics, Chobanian & Avedisian School of Medicine, Boston Universit, Boston, Massachusetts, United States of America, 4 Department of Chemistry, School of Applied Science and Humanities,Vignan's Foundation for Science Technology and Research, Guntur, Andhra Pradesh, India, 5 Department of Medical Laboratory Technology, Faculty of Applied Medical Sciences, University of Tabuk, Tabuk, Saudi Arabia, 6 Prince Fahad bin Sultan Chair for Biomedical Research, University of Tabuk, Tabuk, Saudi Arabia, 7 Centre for Bioinformatics, Department of Biochemistry, Karpagam Academy of Higher Education, Coimbatore, Tamil Nadu, India, 8 School of Pharmaceutical Science and Technology, Tianjin University, Tianjin, China, 9 Department of Medical Microbiology, Tigray Health Research Institute, Mekelle, Tigray, Ethiopia

‡ First Author(s).
* ggoyitom@yahoo.com (GGG); paradess@srmist.edu.in (PD)

## Abstract

Canonical Wnt signaling plays a key role in tumor cell proliferation which correlates with the accumulation of β-catenin resulting inactivation of the network of targets such as GSK3β, Axin, CK1. Uncontrolled expression of β-catenin leads to different types of cancers and other diseases such as sarcoma and mesenchymal tumor formation. However, β-catenin is an attractive target for cervical cancer. In the present study, the compounds such as Doxorubicin and Zinc conjugated with Doxorubicin were screened against β-catenin using Molecular Docking, Molecular Dynamics Simulation, MM/GBSA, and DFT approaches to explore their insights. The study further demonstrated that the binding energy of Zn conjugated with Doxorubicin has shown -7.2 kcal/mol and Doxorubicin registers -5.9 kcal/mol against β-catenin. The disruption between the β-catenin/Tcf-4 complex was observed through the Zinc-Doxorubicin complex, both the proteins are separated about 12 Å. The Zn-Doxorubicin was stabilized with the hydrophobic residues such as Val349 of β-catenin and Phe21 of Tcf-4. The DFT analysis using the B3LYP/6-31g(d,p) method explores that Zn-doxorubicin in complex with the binding site residues has shown the HOMO-LUMO gap of 2.55 eV. The binding free energy calculations exhibit the Zn conjugated Doxorubicin favors in the study by showing ~3 kcal/mol difference with Doxorubicin. The Zn-conjugated Doxorubicin will be discussed in the context of cervical cancer with the hope of improving drug efficacy and reducing toxicities for the betterment of the patient's quality of life.

**Data availability statement:** All relevant data are within the manuscript and its Supporting Information files.

**Funding:** The author(s) received no specific funding for this work.

**Competing interests:** The authors have declared that no competing interests exist.

## Introduction

Cervical cancer is a major worldwide health issue that leads to elevated levels of illness and death. Roughly half a million women have cervical cancer annually [1]. Significant instances are documented in underdeveloped nations lacking efficient screening methods. Risk factors include the following: exposure to immune system dysfunction, human papillomavirus, smoking, genetic effects, and viral infections [2]. Cervical cancer arises from cervical intraepithelial neoplasia (CIN), a condition in which the presence of human papillomavirus (HPV) infection is crucial for the formation of abnormal cervical tissue [3]. In addition to HPV infection, several host or environmental variables can potentially contribute to the development of a malignant phenotype. The primary significant co-factors include the β-catenin protein, which is a crucial element of the Wnt/β-catenin signaling system. β-Catenin plays a crucial role in two main biological processes: adhesion between cells and the transmission of signals [4]. Recent reports have demonstrated a link between the human papillomavirus (HPV) protein and the longer non-coding RNA (lncRNA) in the development of cervical cancer [5].

The Wnt signaling pathway has a significant impact on the process of malignant cells de-differentiation and proliferation [6]. Each tumor contains subclones that exhibit a wide range of gene alterations and promoter hypermethylation in each cells that makes up the tumor and its surrounding environment [7–9]. The mechanism indicates that the primary driver of tumor growth is defective Wnt pathway activation [10]. Tumor formation occurs as a result of the l Wnt signaling pathway, which results in the excessive buildup of β-catenin in the nucleus due to the deactivation of Glycogen Synthase Kinase-3 (GSK-3). Disruption involving Wnt signaling by means of many methods contributes to the development of cancer, with over 95% of cancers being caused by mutations in β-catenin [11].

The primary mechanism of Wnt signaling regulation involves the degradation of β-catenin. This process starts with the phosphorylation in β-catenin by GSK-3, followed by ubiquitination and proteolysis through the means of proteasome [12]. Initiation of the Wnt signaling pathway causes β-catenin to diffuse into the nucleus, which then leads to the activation of target genes as well as attracts T cell factors/lymphoid enhancer factor (TCF/LEF) transcription factor family [13]. Process becomes unstable as a result of the inactivation of the destruction complex, leading to a continual provision of non-phosphorylated β-catenin to nucleus [12], which ultimately causes the excessive expression of genes (Fig 1).

β-catenin functions as a co-activator when it forms a complex together with transacting Tcfs or LEF-1. This interaction is primarily responsible for the development of many types of tumors called carcinomas [15]. The disruption of this complex, which delays the transcriptional activation of β-catenin/Tcf target genes, is regarded as an essential therapeutic treatment. The dysregulation of transcription in cancer is caused by the overactive β-catenin, which hinders the building up of the β-catenin/Tcf complex [13]. This can occur through both blocking the non-phosphorylated β-catenin or through incorporating a competitive inhibitor for Tcfs, thus disrupting their binding with β-catenin [16]. Chemotherapy is a commonly recommended treatment for managing the development of cancerous tumors. Nevertheless, the lack of patient response to treatment while on chemotherapy or the reappearance of cancer following treatment is a significant challenge that greatly impacts the effectiveness of chemotherapy. This resistance is due to genetic differences, especially in tumoral somatic cells. [17,18]. Artem Blagodatski et al., reported that a potent compound called β-elemene, derived from an ancient Chinese medicinal herb called *Curcuma zedoaria*, has demonstrated the ability to inhibit the growth and movement of SiHa cells, a type of cervical cancer cell type. This compound achieves this by reducing the levels of β-catenin and Tcf7 [16]. A groundbreaking finding by Roberts et al., 2017 [19]. has led to the development of a new

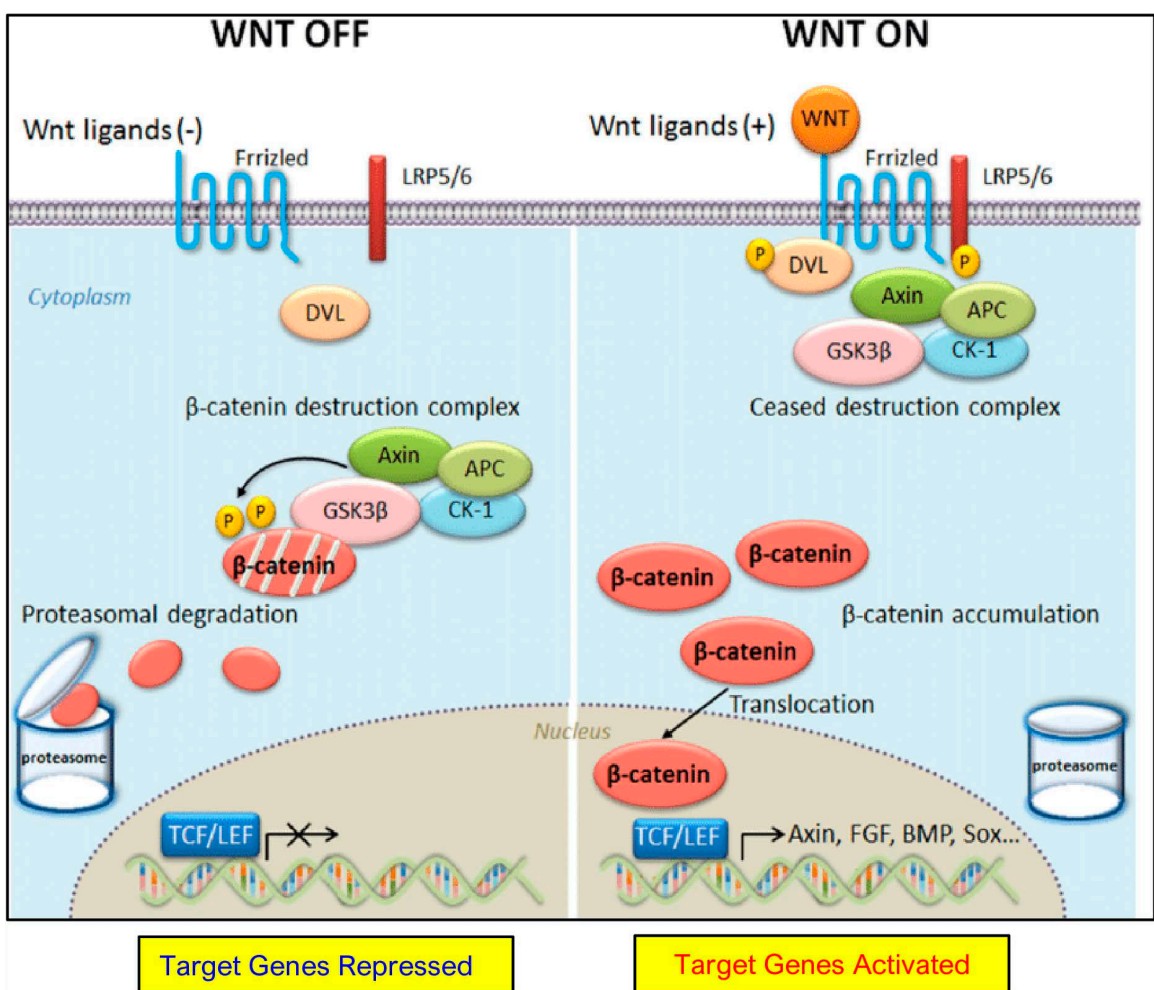

**Fig 1. A comprehensive examination of the Wnt/ β-catenin signaling pathway.** Eliminating the presence of Wnt ligands ("WNT OFF"), the destruction complex phosphorylates cytosolic β-catenin, leading to its recognition and subsequent proteasomal breakdown. When Wnt ligands are present ("WNT ON"), The main purpose of the "destruction complex" is prevented in the intent to phosphory-late cytosolic β-catenin. The introduction of β-catenin that is not phosphorylated in the cytosol causes it to go into the nucleus, which results in the production of Wnt target genes which results in the T-cell factor along with lymphoid enhancer factor-1 (TCF/LEF1) family associated with transcription factors [14].

compound that effectively targets β-catenin as well as its co-activator BCL9. This compound has shown promising results for decreasing cholesterol homeostasis and inhibiting the growth of Colorectal Cancer cell lines [20]. The inhibition of β-catenin/BCL9 complexes should not affect the network with the other partners such as E-cadherin, Axin and APC which results in reducing toxic side effects [10,21]. Recently, Yang et al., 2021 [reference] listed the beta-catenin inhibitors in their review. They have listed the natural and other compounds based on HTS. Collectively, the compound PKF118-310 and ZTM0000990 with the $IC_{50}$ of 0.8 and 0.64 respectively. Lepourcelet et al., has proposed these compounds prevents the interaction between Beta-catenin and TCF4 in colon cancer and destroy the β-catenin/ACP complex.

Doxorubicin (DXR) is a part of the anthracycline family and this drug is commonly used chemotherapeutic drug to treat all types of cancer [22]. DXR may cause drug resistance and tumor growth which leads to poor patient prognosis and survival [23]. Signaling path-way interaction can aid DXR resistance through the induction of proliferation, cell cycle

progression, and apoptosis prevention [23,24]. The resistance may also occur due to the efflux of cancer cells overexpressing the multidrug resistance protein or P-glycoprotein. Earlier reports demonstrated that the DXR is resistance against several cancer targets but still it remains an unsolved issue in the cancer treatment with resulting side effects [24]. Pelin Mutlu et al., 2018 reported the changes in the expression levels of 186 genes leads to drug resistance. Where the Wnt signalling pathway comprises of doxorubicin sensitive and HeLa cell line (36 genes) and K362 cell line (17 genes) through qPCR method. Further, the obtained experimental values shown that 0.031 $\mu$M against K562 cell line and 2.664 $\mu$M against HeLa cell lines. However, the molecular mechanism of Doxorubicin in cervical cancer cells remains unclear. Considering the variation of cancer phenotypes among human individuals, the response of drug conjugate could be efficacious across all the individuals. The result of the previous work indicates that the binding affinity has been improved significantly between the protein-ligands against Syk and RSK2 target of breast cancer [25,26].

Metals and metal oxides are excellent choices due to their extensive range of oxidation states, versatility, and unique chemical properties [27]. Notably, the US Food and Drug Administration (FDA) has accepted zinc as one of the five metal oxides considered safe [28]. Under acidic conditions, the $Zn^{2+}$ ions dissociate quickly, exposing their cytotoxic properties [29]. The tumor is enveloped by an acidic environment that regulates the release of chemicals at the cancerous site through the use of acidic pH-responsive Zn, while keeping the healthy cells undisturbed. $Zn^{2+}$ has a significant effect on various cellular processes, including mitochondrial dysfunction, oxidative stress, lipid peroxidation, reactive oxygen species (ROS) and also DNA damage, ultimately resulting to cell death [30]. Regarding this case, the $Zn^{2+}$ helps facilitate the adsorption of small organic molecules, thereby boosting their activity. Understanding the major interactions, such as Coulombic, hydrophobic, or covalent, provides an extensive insight into how Zn-drug conjugates can effectively target malignant cells and induce apoptosis [31].

Understanding the complex interaction network pathways between target and drug is crucial. Hence, the computer-aided drug discovery approach provides more insights to showcase the pharmacological network between the targets. To our knowledge, investigation of $\beta$-catenin occurring with doxorubicin and zinc conjugated with doxorubicin has not yet been reported elsewhere. Speculating the investigation of the binding of Doxorubicin and Zn conjugated Doxorubicin to the Tcf-binding sites of $\beta$-catenin aims to inhibit the interaction between Tcf and $\beta$-catenin along with the Wnt/$\beta$-catenin signaling pathway, which brings about decrease in transcription of the target genes that are not essential. The purpose of the current study was to use computational methods such as Molecular Docking, Molecular Dynamics and DFT approaches to comprehend the importance of doxorubicin and Zn-doxorubicin, which may provide a novel strategy into the Wnt/$\beta$-signaling pathways on cervical cancer treatments.

## Experimental procedure

### Ligand preparation

Firstly, the existing compounds PKF118-310 and ZTM0000990 has been reported by Yang et al., 2021. The $IC_{50}$ of the above-mentioned compounds are 0.8 and 0.64 $\mu$M respectively. The doxorubicin structure was extracted from drug bank (www.drugbank.ca) and the Zn was conjugated with doxorubicin, all the taken compounds were re-drawn using Cambridge Soft ChemDraw Ultra *Ver*. 9.0 [32] (Fig 2). The LigPrep module of Schrodinger was employed to prepare the ligands [33]. The structures were processed using the default parameters along with the OPLS-2005 force field and all ligands were maintained at pH of 7 ± 2.

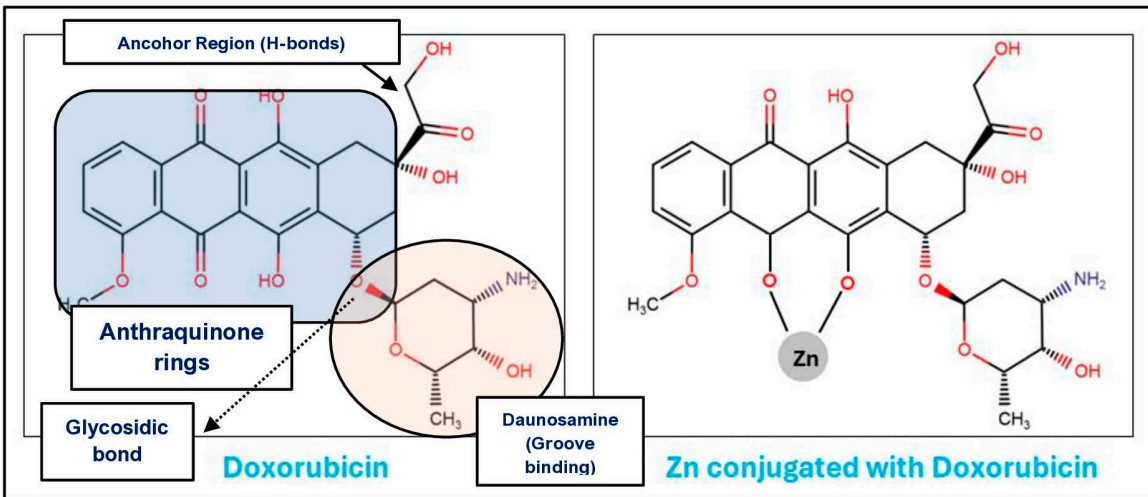

**Fig 2. Structure of Doxorubicin and Zinc conjugated with Doxorubicin.**

## Protein preparation and receptor grid generation

The structure of β-catenin (PDB ID: 1JDH) was extracted from the protein data bank (PDB) (http://www.pdb.org/) [34]. The protein, consisting of A and B chains and comprising 529 amino acids, was prepared by assigning appropriate bond orders and hydrogens to generate the relevant ionization states at pH 7.0. Missing residues and formal charges on ionizable residues were added. The structure was then optimized using the OPLS-2005 force field and minimized to eliminate steric clashes. The SiteMap module of Schrodinger was employed to predict the binding site regions [35]. The grid was generated using Receptor Grid Generation module of the Schrodinger and it was extending for molecular docking analysis. With no constraints defined, the ligand remained flexible for docking with the receptor. The receptor atoms were scaled with a Van der Waals radius factor of 1 Å and a partial cut-off of 0.25 Å. Further, the molecular docking calculations were carried out using Glide-XP [36,37].

## Molecular dynamic (MD) simulations

The molecular docking outcome (β-catenin – Doxorubicin and β-catenin – Zn conjugated with Doxorubicin) was used as a starting structure for MD simulation. The calculations were performed using the Desmond molecular dynamics package of the Schrodinger Suite [38]. using the OPLS-2005 force field. The box dimensions ensured of containing the protein complexes of 10 x 10 x 10 Å with periodic boundary conditions and solvated using Simple Point Charge (SPC) water molecules. The complex was neutralized by adding Na + and Cl- ions. Further, the energy minimization step was carried out using the Steepest Descent gradient algorithm for 2000 steps with a convergence threshold of 1.0. Noose-Hoover chain and Martyna-Tobias-Klein barostat [39]. were used to keep the system in a stable environment (300k, 1 bar) and the relaxation time of 2ps and the electrostatic cut-off was set to 9 Å. The equilibrated complex was carried out for 100 ns simulation followed by recording at an interval of 50ps [40]. The simulation for both the complexes was performed as triplicates and the study demonstrates the average interaction shown by the binding site residues.

## MM/GBSA analysis

The MM/GBSA of prime module of Schrodinger was employed to predict the Binding Free Energy between the protein and ligand. The calculation was performed for every 10th frame of molecular dynamics simulation of the protein-ligand complex. The binding free energy was calculated using the following equation

$$\triangle G_{bind} = \triangle E + \triangle G_{solv} + \triangle_{SA} \tag{1}$$

## DFT calculations

DFT calculations were carried out using the Gaussian 09 [41]. program to determine the hydrogen bonding strengths, molecular orbitals (HOMO-LUMO), and binding energy between the ligand and selected amino acids [42]. Owing to the computational limitation, we have considered only closely interacting amino acids with the Zn-doxorubicin and doxorubicin for the DFT study. Graham et al., 2001 described that human Tcf4 is the predominant form in the color cancer cell lines [43]. Therefore, Tcf4 interacting residues were considered (Arg386, Asp459, Asn415, Thr418) to calculate the binding energy with the Doxorubicin and Zn-Doxorubicin. Additionally, a DFT study was conducted to understand the interactions between key amino acids in the binding pocket and the ligands at the molecular level. The geometries are optimized at B3LYP/6-31g(dp), M06/6-31g(dp), BHandHLYP/6-31g(d,p), and wB97XD/6-31g(d,p) level of theory in gas phase. Molecular orbitals are visualized using Gauss view 5.0.8 [41]. The binding energies are calculated using the following equation:

$$\text{Binding Energy} \left(BE\right) = E_{\text{Ligands} - \text{Amino Acids}} - \left(E_{\text{Ligands}} + E_{\text{Amino Acids}}\right) \tag{2}$$

## Results and discussions

The STRING database was employed to understand the interaction analysis of β-catenin protein [44] The metabolic cascade demonstrates the interaction network of β-catenin with other targets (Fig 3). The STRING database deals with array of approaches to provide detailed insights to access both experimental and predicted interactional data. The interaction scores in STRING database expresses approximate confidence (between 0 and 1) of the association being true, provided with the experimental evidence. Here, the clustering model k-means of clustering was utilized to detect the closed B-catenin protein targets. The outcome shows that region indicated in cluster-1 is closer when compared to the cluster-2 and cluster-3.

### Residual Interactions in the absence and presence compounds

Graham et al., 2001 reported the interaction between β-catenin and Tcf4 [43]. The interaction between the two domains were mediated by 42 amino acid sequence motif or armadillo repeat. The reported three binding modules are (i) β-hairpin module in N-terminus, (ii) extended central region found in β-strand conformation and (iii) C-terminal α helix. Further, results manifested that charged interaction (negative–positive) provides significant binding energy with lower entropic penalty. Hence, we have measured the distance between Lys312 and Glu24 was found to be 8.5 Å before the docking. Whereas the distance extended to 12 Å after binding with Zn-Doxorubicin and the complex was narrowed to 7.5 when the Doxorubicin binds. Hence, this analysis explores that the Lys312–Glu24 interaction is crucial for binding (Fig 4).

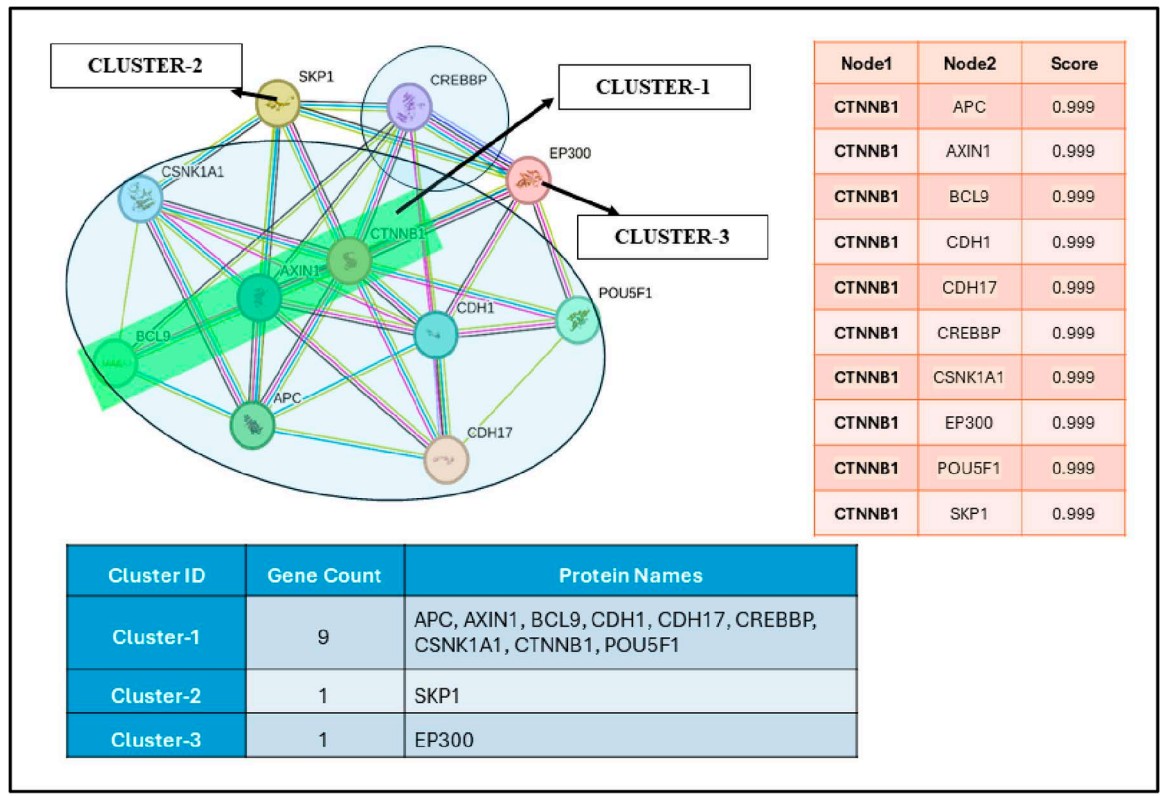

| Node1 | Node2 | Score |
|-------|-------|-------|
| CTNNB1 | APC | 0.999 |
| CTNNB1 | AXIN1 | 0.999 |
| CTNNB1 | BCL9 | 0.999 |
| CTNNB1 | CDH1 | 0.999 |
| CTNNB1 | CDH17 | 0.999 |
| CTNNB1 | CREBBP | 0.999 |
| CTNNB1 | CSNK1A1 | 0.999 |
| CTNNB1 | EP300 | 0.999 |
| CTNNB1 | POU5F1 | 0.999 |
| CTNNB1 | SKP1 | 0.999 |

| Cluster ID | Gene Count | Protein Names |
|-----------|-----------|--------------|
| Cluster-1 | 9 | APC, AXIN1, BCL9, CDH1, CDH17, CREBBP, CSNK1A1, CTNNB1, POU5F1 |
| Cluster-2 | 1 | SKP1 |
| Cluster-3 | 1 | EP300 |

**Fig 3. Protein Interaction Network of CTNNB1 ( β-catenin) using String database.**

## Molecular docking analysis

The docking of Doxorubicin and Zn-Doxorubicin derivatives into the binding site of the Tcf4 - β-catenin protein (PDB ID: 1JDH) was performed using Schrödinger software, to compute ligand conformation and orientation relative to the active site of the target protein. Five binding sites were predicted through the SiteMap analysis. The molecular docking was carried out for all the five predictions. Out of which, the third prediction has shown the significance in the binding when compared to other.

Yang et al., 2021 has comprehensively listed the β-catenin inhibitors along with their experimental values in their review. We have started the screening procedure with the two β-catenin inhibitors such as (i) PKF118-310 (IC$_{50}$: 0.8 μM) and (ii) ZTM000990 (IC$_{50}$: 0.64 μM) as a reference (S1 a&b Fig). The PKF118-310 has shown the binding energy -4.13 kcal/mol and ZTM000990 has shown -8.50 kcal/mol (S1c Fig). The ZTM000990 has taken as a reference compound for our study.

To understand their structural insights, the molecular docking analysis was performed for Zn-Doxorubicin and Doxorubicin with β-catenin. The interaction of Zn-Doxorubicin and Doxorubicin shows that binding energy value of -7.2 kcal/mol and -5.9 kcal/mol, respectively. The Zn-Doxorubicin has shown hydrogen bonding interaction with Phe21 and Glu24 of Tcf-4 and other residues such as Lys354, Val349, and Arg386 of β-catenin and represented in Table 1. The residue Val349 accommodates the Zn-doxorubicin inside the hydrophobic pocket. The Doxorubicin shows a hydrogen bonding interaction with the amino acids such as Glu28 of Tcf-4 and Trp383, Lys345, Trp383 and Asn415 of β-catenin

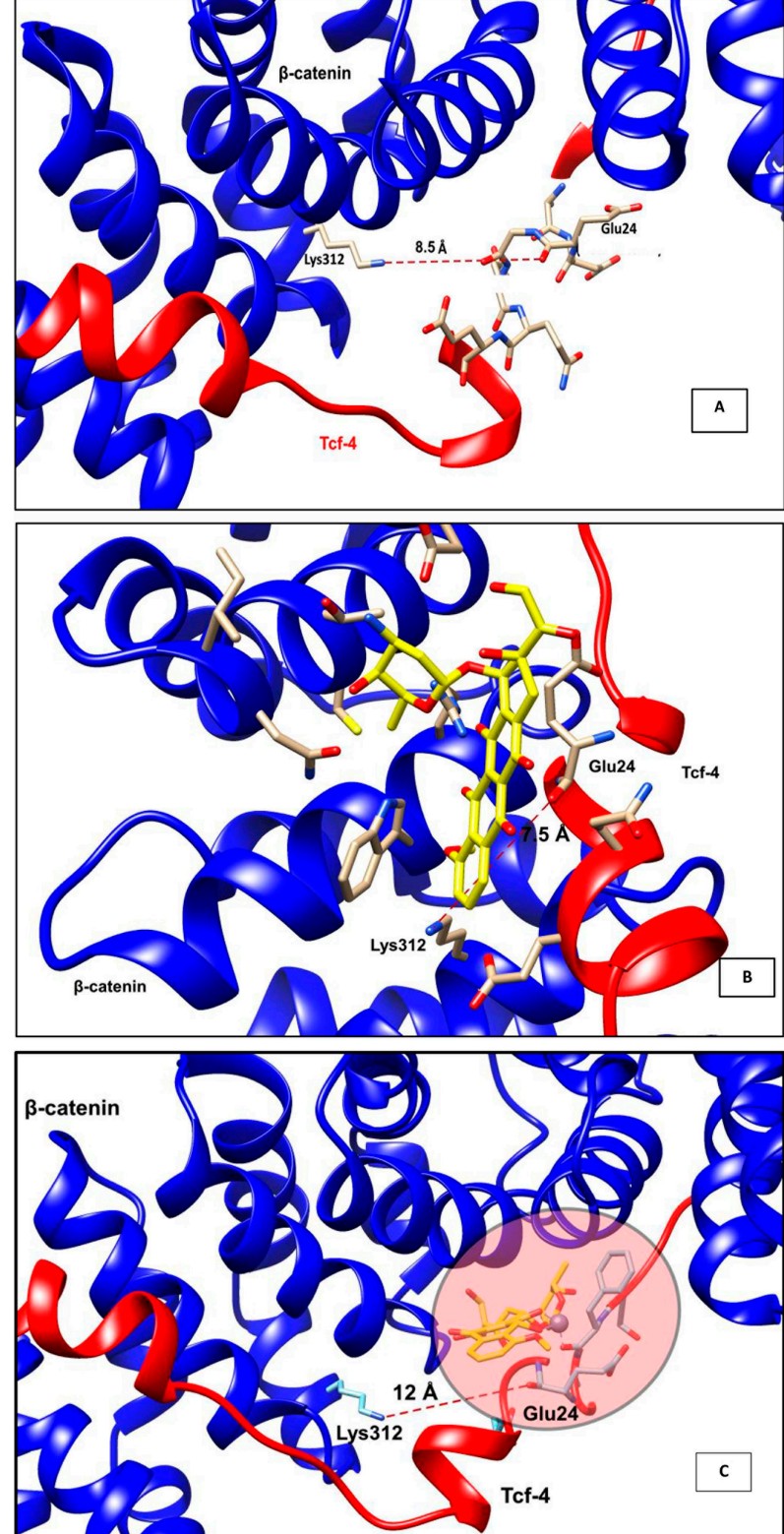

**Fig 4. Distance analysis between Lys312 of β-catenin and Glu24 of Tcf-4** (A) prior to docking (B) presence of absence of Doxorubicin and (C) after binding with Zn-Doxorubicin.

(Fig 5). The interaction pattern of Doxorubicin and Zn conjugated with Doxorubicin were found to be similar. The salt bridge interaction was noticed between Asp459 and Doxorubicin and the π-π interaction were observed with Trp383. The Zn conjugated with Doxorubicin also shows salt bridge interaction with Asp390 and π-π interaction was noticed with Trp383. The conformation change occurred when the Zn conjugated with Doxorubicin. Hence, the study indicates that Zinc ion plays a significant role to prevent the contact between the two complexes.

## Molecular dynamics simulations

Molecular dynamics simulations of the protein-ligand complex were performed using Desmond, Schrodinger, LLC, NY, USA for 100 ns. The study aimed to understand the stability of the complex in the solvated system. Hence, the simulation was triplicated and considered the doxorubicin as a reference compound against β-catenin to examine the stability of Zn-Doxorubicin.

The MD simulation analysis reveals that the doxorubicin-β-catenin complex converged immediately after the equilibration. The RMSD plot shows that the protein has raised up to 9 Å resulting that there would be conformational change. As the region is surrounded by water molecules, the ligand may have flexibility and show various interactions. Hence, it may be the result of a large gap (S2 Fig). Whereas the Zn-Doxorubicin bounds with β-catenin was converged after the complex equilibrated and raised up to 6 Å and the ligand maintains within 3 Å. The Zn-Doxorubicin enclosed in the hydrophobic pocket containing the residue Phe21. (S3 Fig). The larger RMSD gap indicates instability, whereas the smaller gap in RMSD leads to stability [45]. Hence, the Doxorubicin protein complex has shown a large gap as it may exhibit a delayed response to converge, whereas instantaneous convergence resulting in the complex is more favorable in the case of Zn-Doxorubicin.

Fig 6 represents the average interaction of the Doxorubicin β-catenin complex where it mainly has contacts with Glu24, Arg386, Asp459, Glu462, Lys22, and less interaction with

**Table 1. Angles and Hydrogen bond measurements for β-catenin/Tcf-4 complex.**

| Complex | Glide XP Score (kcal/mol) | Angle | Distances |
|---|---|---|---|
| ZTM000990 | -8.50 | Glu29 C-O...O – 135.9° | Glu29 OD1 – O – 2.8 Å |
| | | Val349 C – O... O – 125.1° | Val349 O – O8 – 2.9 Å |
| | | Val349 C – O... O – 124.3° | Val 349 O – O6 – 2.9 Å |
| | | | Lys30 N – O2 – 2.7 Å |
| | | | Lys30 N – O4 – 3.2 Å |
| Zn-Doxorubicin | -7.20 | Asp390 C – O... O – 102.5° | Asp390 – OD1 – O4 – 2.7 Å |
| | | Phe21 C – O... Zn – 120.6° | Phe21 O – Zn – 2.6 Å |
| | | Zn-C19...O9 – 127.1° | Phe21 N – O2 – 3.0 Å |
| | | Zn-C12...O5 – 120.4° | Zn - O5 – 1.82 Å |
| | | Val349 C-O...O7 – 89.5° | Zn - O9 – 1.80 Å |
| | | | Val349 O – O7 – 3.0 Å |
| | | Lys354 CE – N +... O - 125° | Lys354 N – O – 3.0 |
| | | CZ – NH1 – O11 – 137° | Arg386 NH1 – O11 – 2.9 Å |
| Doxorubicin | -5.90 | Asn415 CG – OD1... O4 - 130° | Asn415 OD1 – O4 – 3.2 Å |
| | | Trp383 CD1-NE1-O2 – 111.2° | Trp383 NE1 – O2 – 2.7Å |
| | | Lys345 CE-N +... O2 – 113.1° | Lys345 N – O2 – 2.7 Å |
| | | Glu28 CD-OE1-O8 – 118.7° | Glu28 O – O8 – 2.6 Å |

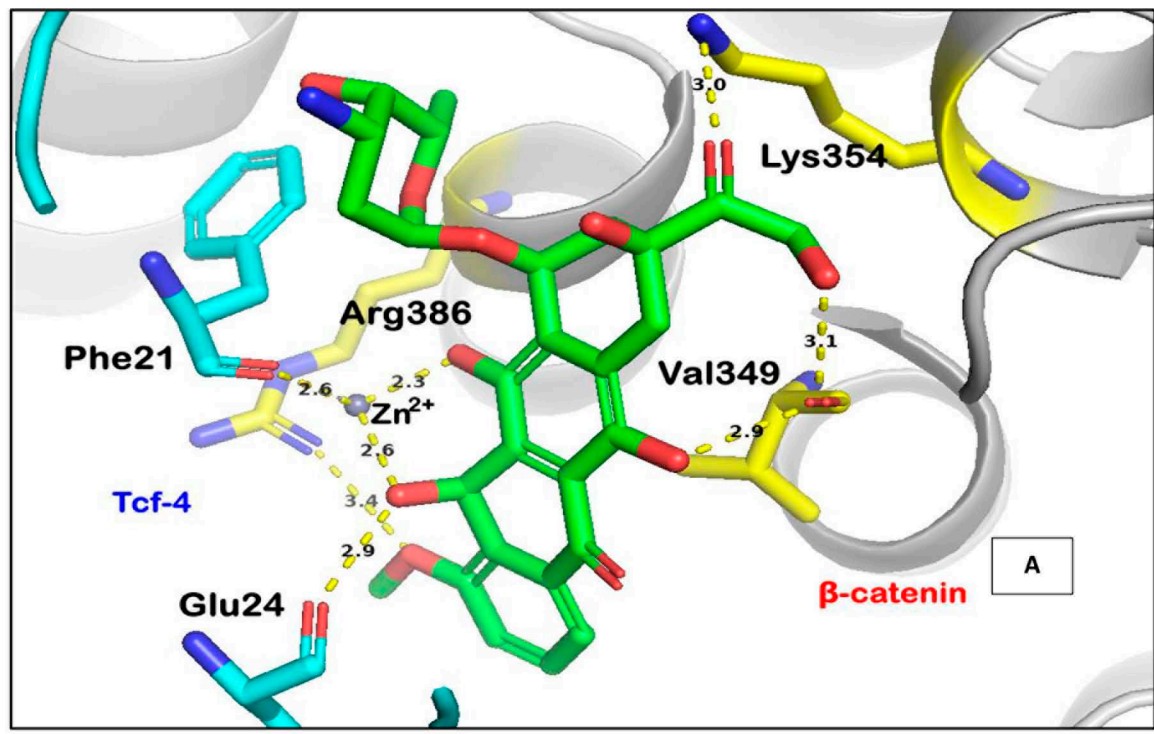

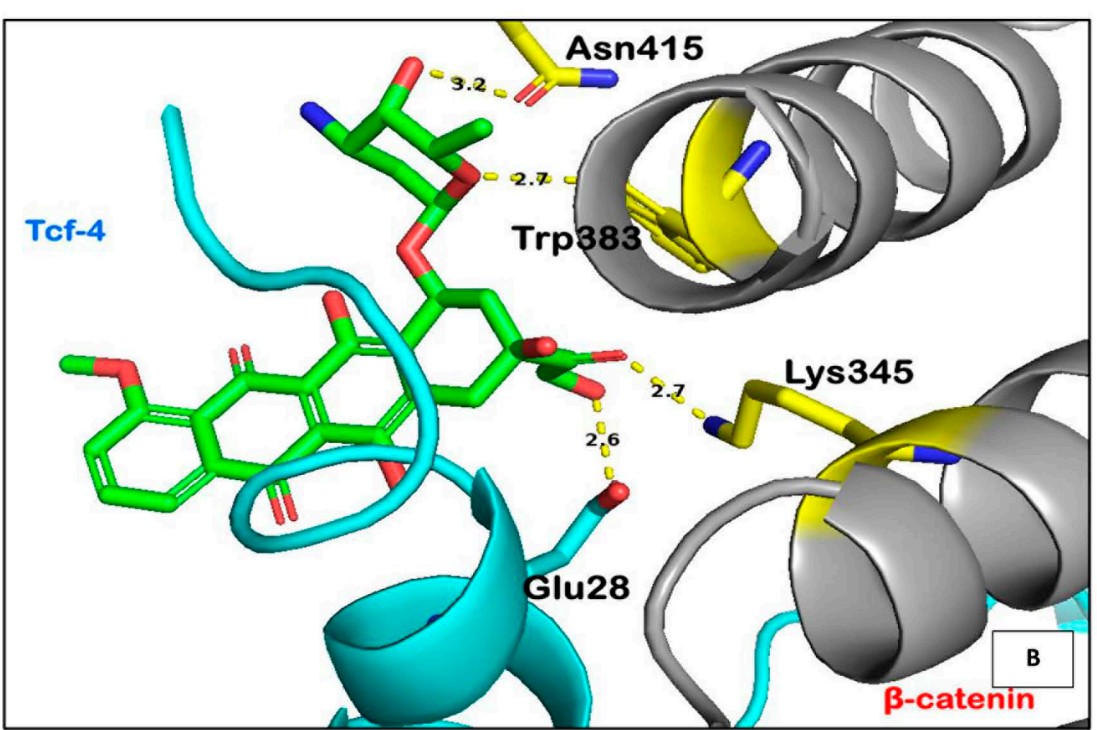

**Fig 5. Interactions of** (A) Zn-Doxorubicin–1JDH (B) Doxorubicin–1JDH. In both (a) and (b), the Trp383 shows pi-Alkyl and π-π stacked interactions.

Phe21. Whereas the water molecule stabilizes the complex along with Arg386. Also, in the anchor region of doxorubicin was covered about ~ 70% of water molecules, channels with the charge residue Glu28. The daunosamine region was engulfed by hydrophobic residue Trp383 and polar residue Asn415. Whereas the anthracycline core region is covered mainly with the water and Glu24 residue. Hence, it is finding the place to stabilize in the complex. On the other hand, **Fig 7** represents the Zn conjugated with Doxorubicin complex of average contacts mainly with hydrophobic residues such as Leu18, Phe21, Val349, Trp383 including the ionic contact with Phe21 and Glu24. Precisely, the Zn-Doxorubicin-β-catenin complex has shown interactions mainly with the hydrophobic residues such as Phe21 and Ile19 with duanosamine region. The anchor region shows interaction with Ser348 and Lys354 where it is surrounded by Val349. The water molecule did not appear near the anthracycline core of the Zn-Doxorubicin-β-catenin complex, where the Phe21 and Trp383 are facing toward each other.

## MM/GBSA analysis

The Binding Free Energy was computed every $20^{th}$ frame of the protein-ligand complex, and the average value was reported. The ZTM000990 has shown an average value of -52.84 kcal/mol followed that the Doxorubicin and Zn-Doxorubicin resulted in -48.25 kcal/mol, -57.36 kcal/mol, respectively. The study suggests that Zn-Doxorubicin has shown better interaction with the Val349, Lys354, and Arg386; Phe21 of Tcf-4. In contrast, the Doxorubicin has shown interaction with charged residues and less interaction with Phe21 (hydrophobic residue). The $dG_{coulomb}$ for ZTM00990, Zn-Doxorubicin and Doxorubicin has shown -39.26, -68.93

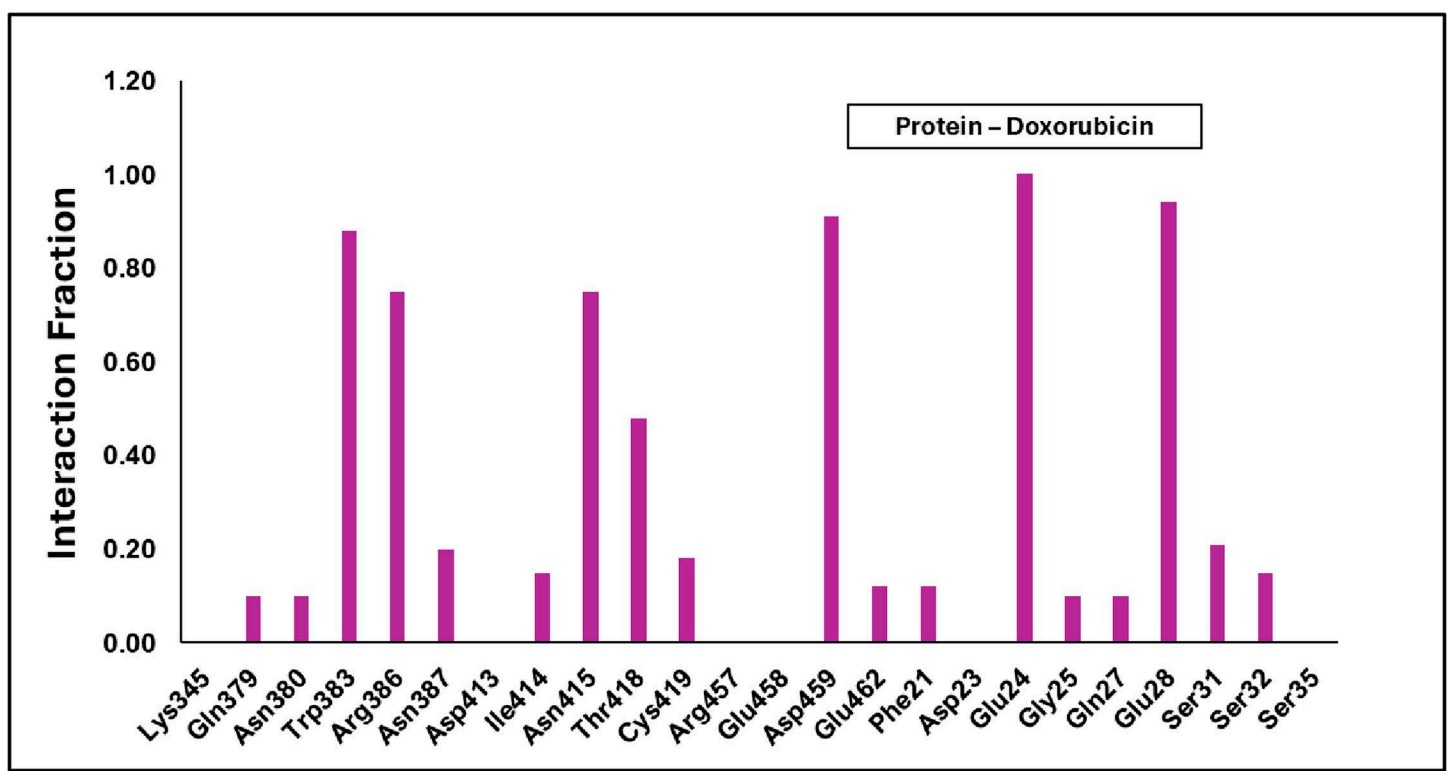

**Fig 6. Average interaction of β-catenin and Doxorubicin contact at the binding site.** The region is stabilized in the presence of water molecules and the charged residues.

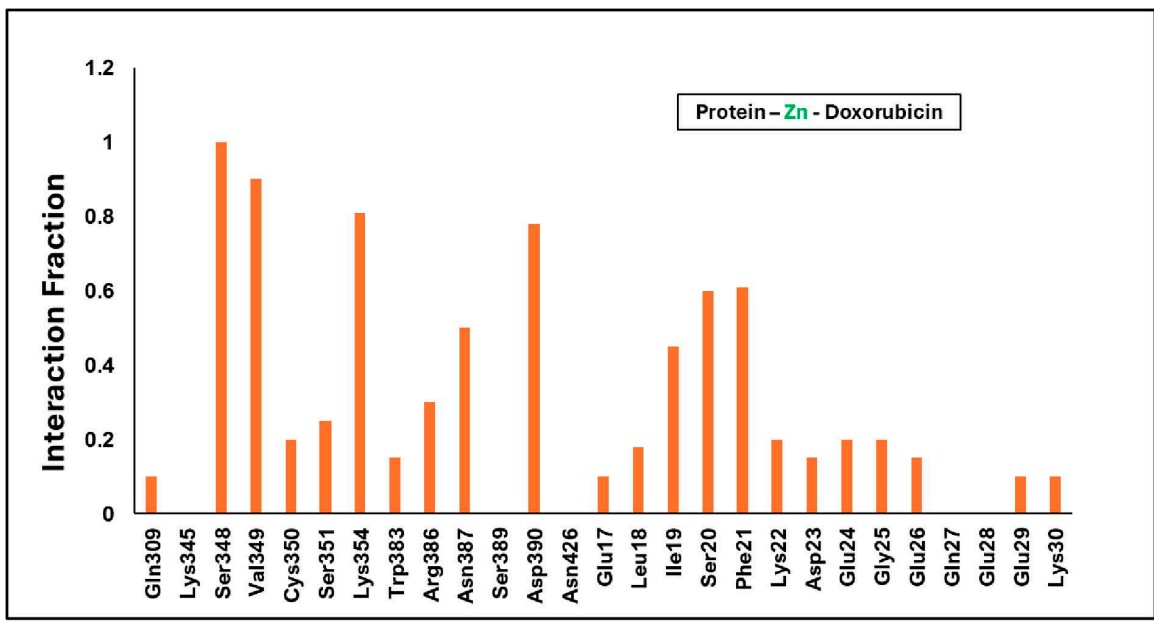

**Fig 7. Average Interaction of Protein-Zn-Doxorubicin Complex.** The region is stabilized with the hydrophobic residues such as Phe21 and Ile19.

and -35.66 kcal/mol respectively (Table 2). This study shows that interaction energy between the protein and ligand is found optimized in the presence of $Zn^{2+}$ and due to the presence of hydrophobic residues in the binding site, hence, the complex was identified as stable (S4 and S5 Fig). The lowest energy of $\Delta G_{bind}$ values indicates the compactness of the ligands inside the binding pocket of β-catenin.

Solvation and desolvation effects are considered a most effective approach to evaluate the global and local conformation of protein in the solvent model [46]. The solvation-free energy values were calculated for every residue, and it was averaged. The averaged solvation energy for ZTM-000990 has shown -5.35 kcal/mol, and Doxorubicin and Zn-Doxorubicin complex has shown -3.38 and -6.55 kcal/mol respectively. The difference between the Zn-Doxorubicin and ZTM-000990 has shown ~ 1.0 kcal/mol order of magnitude whereas the Doxorubicin has shown ~ 3 kcal/mol. However, the $Zn^{2+}$ has a role of altering the conformation as well as attracting possible contacts to maintain stability.

## DFT analysis

DFT calculations were performed on the ligands that achieved the highest scores in the docking studies. For DFT studies, the amino acid residues such as Arg386, Asp459, Asn415, and Thr418 have been considered which are shown to be interacting with the ligands observed in

**Table 2. The relative binding energies obtained by MM/GBSA.**

| Complex | $\Delta G_{coulomb}$ (kcal/mol) | $\Delta G_{vdW}$ (kcal/mol) | $\Delta G_{bind}$ (kcal/mol) |
|---|---|---|---|
| ZTM000990 | -39.26 | -56.44 | -52.84 |
| Zn-Doxorubicin | -68.93 | -47.08 | -57.36 |
| Doxorubicin | -35.66 | -42.80 | -48.25 |

the docking studies. The calculated binding energies for the Doxorubicin and Zn-Doxorubicin has shown -108.56 and -134.91 kcal/mol, respectively with M06/6-31G(d,p). Three types of hydrogen bonding have been observed in the ligand-amino acid complexes: (i) N-H⋯O, (ii) O-H⋯N, and (iii) O-H⋯O. All hydrogen bond lengths are approximately 2.5 Å, indicating the stability of the complexes (Table 3). The optimized structures of Doxorubicin and Zn-Doxorubicin are displayed in Fig 8.

### Doxorubicin – β-catenin complex

**Zn conjugated with doxorubicin – β-catenin complex.** The binding energies of both the complexes were calculated using DFT methods. The compounds Zn-Doxorubicin and Doxorubicin with the binding site residues have been considered for the DFT computation. The structures were optimized in the following orders (a) only amino acid residues (Arg386, Asp459, Asn415, Thr418), (b) Doxorubicin with given amino acid residues and (c) Zn-Doxorubicin with amino acid residues and it were shown in Fig 9. Frontier MO energies of optimized structures of Doxorubicin and Zn loaded Doxorubicin at various levels in gas phase are presented in Table 4. FMO study has been demonstrated that the HOMO structure of Doxorubicin shows that the electrons are localized on the central part of the ligand. The LUMO structure exhibits that the electrons are localized at Ring B, C and D. The HOMO-LUMO

**Table 3. Binding energy at various level of theory in gas phase for ligand-amino acid complexes.**

| System | Binding Energy (BE) kcal/mol | | | | |
|---|---|---|---|---|---|
| | B3LYP/ 6-31G(d,p) | M06/ 6-31G(d,p) | BH and HLYP/ 6-31G(d,p) | ω97XD/ 6-31G(d,p) | B3LYP-D3/ 6-31g(d,p) |
| Doxorubicin - Complex | -93.66 | -108.56 | -107.56 | -113.64 | -109.92 |
| Zn-Doxorubicin - Complex | -111.35 | -134.91 | -12.69 | -28.08 | -29.65 |

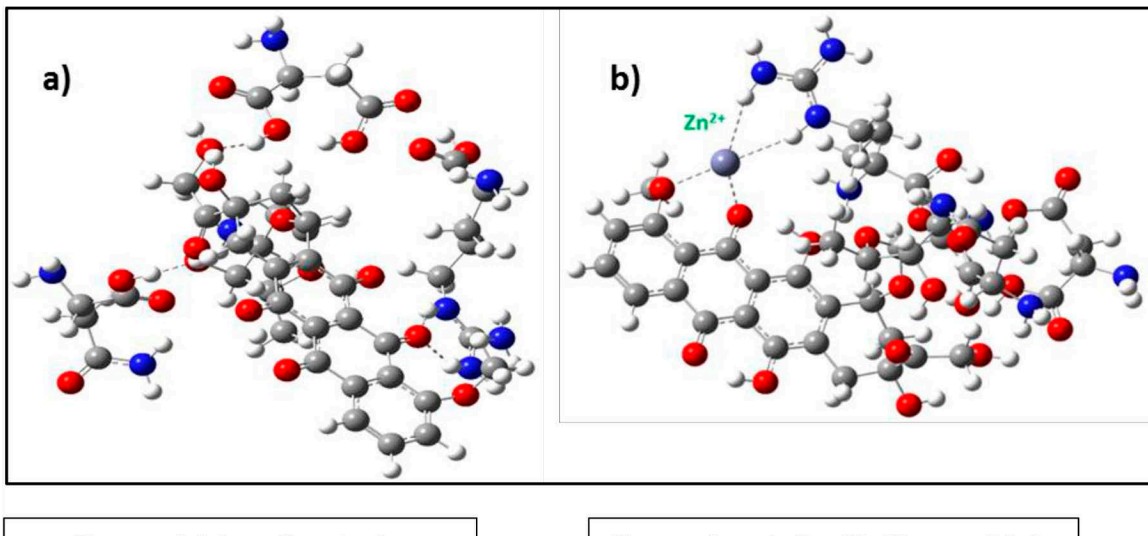

**Doxorubicin – β-catenin complex**

**Zn conjugated with Doxorubicin – β-catenin complex**

**Fig 8.** (a) Optimized structure of Doxorubicin – protein complex and (b) Zn-Doxorubicin bounds with amino acids. Frontier molecular orbital diagram of Doxorubicin and Zn-Doxorubicin with amino acids obtained in gas phase at B3LYP/6-31g(d) level.

analysis indicates that Rings B, C, and D are readily available to donate electrons to the interacting amino acid residues in the protein's binding site. The results show that Zn-Doxorubicin has a higher HOMO energy level compared to Doxorubicin.

The results of DFT studies are in good agreement with the docking results and proves that Zn-Doxorubicin is the potential one against Doxorubicin. The HOMO-LUMO gap, in Zn-Doxorubicin, has been observed to be 2.17 kcal/mol at M06/6-31G(d,p). The study elucidates that the B3LYP/6-31G(d,p) could be a second reliable value because of its consistency. The LUMO of Doxorubicin–complex and Zn-Doxorubicin–complex has scattered from the A ring to B, C, D rings of Doxorubicin, whereas the other two methods BH and LYP and ω97XD has not shown any consistency results in the DFT calculations (S6 Fig). Hence, the present study explores that the M06 method has produced appropriate results.

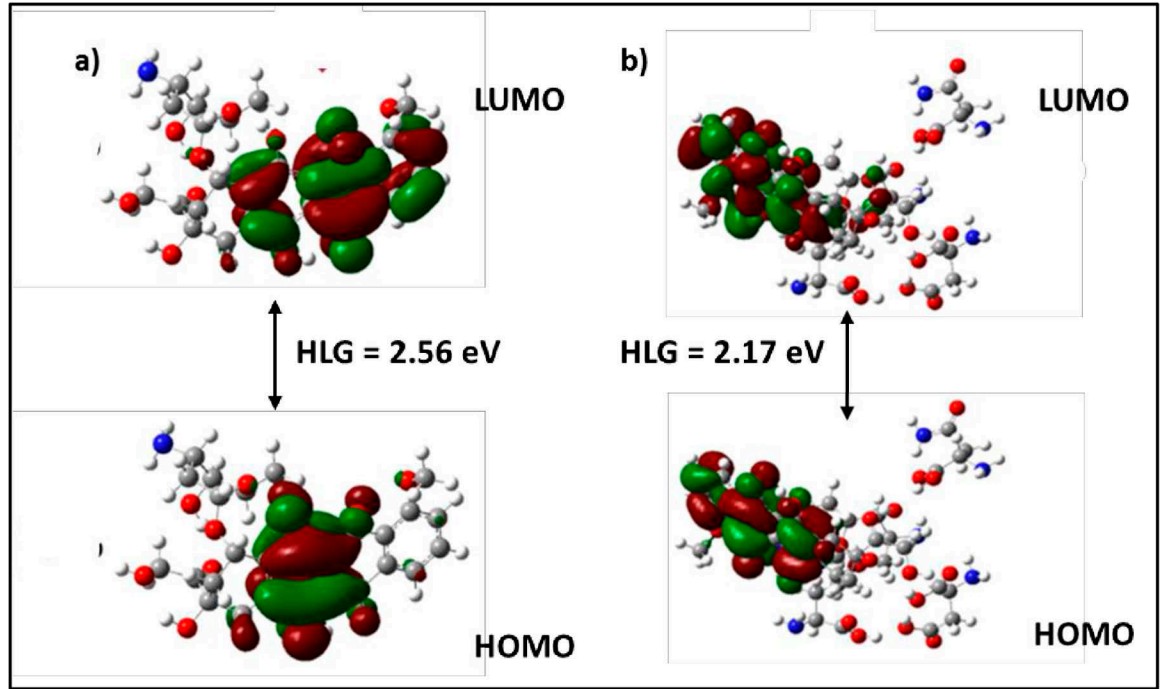

**Fig 9. HOMO-LUMO of** a) Doxorubicin–amino acids (monomer) and b) Zn-Doxorubicin–amino acids (monomer).

**Table 4. FMO energies of optimized structures of Doxorubicin and Zn-Doxorubicin at various levels in gas phase.**

| Methods | Monomer (Arg386, Asp459, Asn415, Thr418) (unit: eV) | Doxorubicin with Monomer (unit: eV) | Zn – Doxorubicin with Monomer (unit: eV)l |
|---|---|---|---|
| B3LYP/ 6-31G(d,p) | 3.27 | 2.59 | 2.55 |
| M06/ 6-31G(d,p) | 3.11 | 2.56 | 2.17 |
| BHandLYP/ 6-31G(d,p) | 5.46 | 4.43 | 4.43 |
| ωB97XD/ 6-31G(d,p) | 6.91 | 4.35 | 5.96 |
| B3LYP-D3/ 6-31g(d,p) | 2.84 | 2.07 | 2.53 |

The lowest HOMO-LUMO gap observed in Zn-Doxorubicin suggests that the HOMO of Zn-Doxorubicin can transfer its electrons to the lower-energy LUMO of the amino acid residues in the enzyme's active site. Consequently, this indicates that the inhibitor's HOMO may transfer electrons to the lower-energy LUMO of the amino acid residues in the active site of the enzyme.

## Discussion

### Conformational changes of the β-catenin/Tcf-4 complex in the presence of $Zn^{2+}$

The protein – ligand complexes are overlaid to understand the conformational changes. The binding position of the doxorubicin and Zn-conjugated doxorubicin were analyzed (Fig 10). The Doxorubicin has shown interaction mainly with the β-catenin and some with Tcf-4. It occupies the comfortable position in other direction and interacts mainly with the β-catenin. Graham et al., 2001 reported that Tcf-4 is the predominant Tcf factor present in colon cancer cells, drugs that disrupt binding between β-catenin and Tcf-4 could be useful in colon cancer. Valerie et al., 2015 [47]. reported that Antipsychotic drug Pimozide (PMZ) inhibits cell growth of Hepatocellular carcinoma (HCC) cell by disrupting the wnt/beta-catenin signaling pathway and reducing Epithelial Cell adhesion molecule (EpCAM) expression. Their study shows that PMZ inhibits cellular proliferation, viability, colony formation ability, and resulting apoptosis. Seung et al., 2017 [48]. has reported the compound HI-B1 disrupts the binding between the β-catenin and Tcf-4. The molecular docking study report mentions that the HI-B1 compound directly interacts with β-catenin especially highlighting the interaction with Lys312. Hence more interactions between the complex will favors the study. In our experiment, the Zn-Doxorubicin binds in the underneath of Tcf-4 and shows reasonable contacts when compared with Doxorubicin-protein complex (Fig 10). To understand the change in conformation, we have extracted the frames periodically and it was superimposed (S4 Fig and

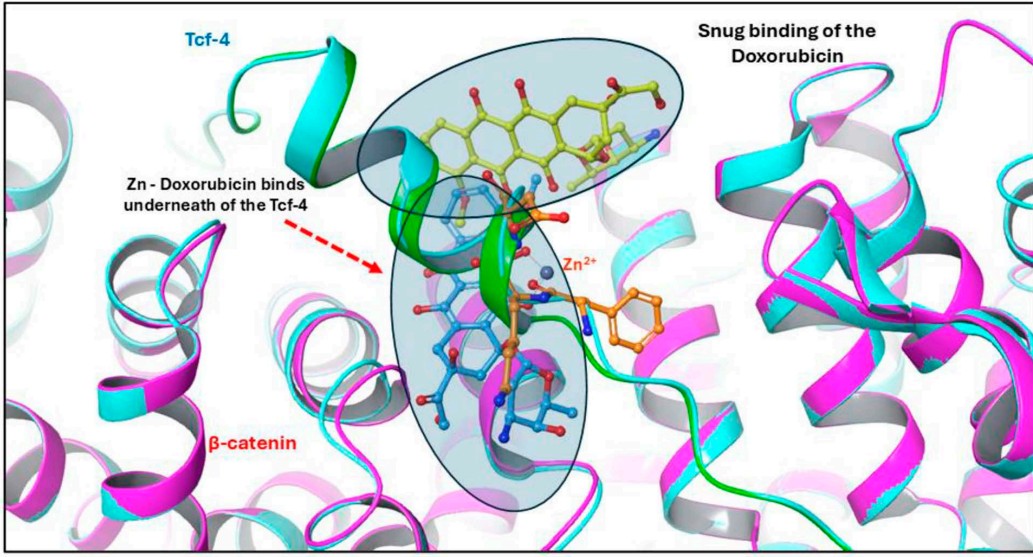

**Fig 10. The docked structure was superimposed to distinguish the binding of Doxorubicin (yellow) and Zn – Doxorubicin (blue).**

S5 Fig), which shows the significance in the different time frames. In the structure analysis, we have observed 1 Å change in the RMSD from the starting structure to the end (100ns). Gao et al., 2020 [49]. studied the molecular mechanism of zinc regulating osteosarcoma. The study shows that zinc treatment significantly inhibited the proliferation and invasion of osteosarcoma cells, leading to cell apoptosis. The zinc activated the Wnt-3a/β-catenin signaling pathway and led to an anti-cancer effect. In the present study, the binding of Doxorubicin against β-catenin/Tcf-4 was found to be other side and showed interaction with β-catenin. On the other hand, Zn-doxorubicin finds a suitable accommodation in between β-catenin/Tcf-4, which may plays a crucial role in the disruption of the β-catenin/Tcf-4 complex, controlling the tumors with higher levels of β-catenin [13,16,19].

Yoon et al., 2024 [50] recalled that zinc plays a crucial role in human health by exploring its anti-cancer, anti-bacterial, and catalytic properties. They experimented with different DFT protocols such as M06 and M06-L functionals to calculate inter-atomic distances between $Zn-(Gly)_2$ and $Zn (Met)_2$. They reported that M06 had produced accurate results by producing the polarization of the two units in the molecular pair for their $Zn-(Gly)_2$ and with a small difference for the rest of their analysis. Hence, our study also demonstrates the M06 functionals produced the highest binding energy of -134.91 kcal/mol and FMO energy of 2.17 eV.

### Functional significance of hydrophobic interactions in the binding site regions

The binding site, which is characterized by hydrophobic interactions, leads to a change in entropy and influences the binding strength [51,52]. The average number of hydrophobic atoms corresponds to one or two donors and three to four acceptors can be observed in the marketed drugs. This highlights the significance of hydrophobic interactions in drug design, as they enhance binding affinity and contribute to binding energies by complementing hydrogen bonding interactions [53]. However, the alternative approach in drug designing is in the presence of water molecules in the place of hydrophobic residues which makes that region flexible and increases the binding affinity.

An increase in the hydrophobic residues in the active core of the drug-target interface leads to produce more drugs showing their biological activity. The functional group plays a crucial role in optimizing the hydrophobic interaction at the protein-ligand interactions. Earlier reports were mentioning that how the hydrophobic interactions will minimize the side effects and toxicity. The study was demonstrated by employing Zn, Cd, Fe, and Mn metal atoms which were docked against c-Src and c-Abl [53]. Further, it was noticed that the ligand containing the metal atom influences the hydrophobic interactions and enhances the binding affinity. The hydrophobic residues play a significant role in the drug-target interface region by strengthening the binding affinity as well as complementing the drug efficacy. However, the present study states that the Zn-Doxorubicin surrounded by the hydrophobic residue Phe21 and Val349 whereas Doxorubicin has no contacts with hydrophobic residues. These interactions suggests that Zn conjugated Doxorubicin could enhance the permeability to overcome the resistance and prevents the growth of β-catenin when compared to Doxorubicin.

### Conclusion

Chemotherapy is one of the efficient treatments for the several cancer diseases. The Doxorubicin remains the effective medication for the Chemotherapy option. However, the drug results many limitations including low solubility, poor bioavailability and many adverse effects.

The conjugated medication treatment shows a path for the effective treatment in the recent scenario. The comparative analysis of Doxorubicin and Zinc conjugated Doxorubicin against β-catenin explore the potencies by showing the binding affinity through the optimized hydrophobic interactions and hydrogen bonding. The result demonstrates that Zinc conjugated Doxorubicin have shown the interaction with the hydrophobic region which is critical to determine the better efficacy of drug leads. This insight which further demonstrates on hydrophobic interactions, hydrogen bonding and metal-associated interactions for drug design. Thus, the present study explores that Zn-Doxorubicin has significant contact with Val349, Lys354 and Asp390 residues which turned as potential for the stability of the complex. Hence, Zinc can penetrate the tumor cells which influences to destroy the cancer cells. The predicted outcomes will provide a foundation for further research to assess the drug-likeness properties of these compounds in inhibiting or blocking the Wnt/β-catenin signaling pathway in cervical cancer. Our findings provide a new perspective for therapeutic targeting of β-catenin.

## Supporting information

**S1 Fig.** (a) Structure of PKF118-310. (b) Structure of ZTM000990. (c) Interaction between protein and ZTM000990. The backbone of Val349 alone shown interaction with the compound. Some regions of the compound are turned off from β-catenin.
(PDF)

**S2 Fig. RMSD plot of Doxorubicin bound to active inhibitory site at β-catenin.**
(PDF)

**S3 Fig. RMSD plot of Zn-Doxorubicin bound to active inhibitory site at β-catenin.**
(PDF)

**S4 Fig. Overlay representation of Protein – Doxorubicin complex.**
(PDF)

**S5 Fig. Overlay representation of Protein – Zn - Doxorubicin complex.**
(PDF)

**S6 Fig. DFT Optimization of the Zn-Doxorubicin complex and Doxorubicin-β-catenin complex in different methods and basis sets.**
(PDF)

## Acknowledgments

GR sincerely thanks Dr. Sudha V, Associate Professor, Department of Chemistry, SRMIST for providing valuable suggestions to carry out the research work. The authors BR and BP acknowledges Alagappa University, Karaikudi for providing computational facilities to carry out this work. The author VSD acknowledges VFSTR for providing Gaussian software to perform DFT studies and also the author SM acknowledge the Research, Development, and Innovation Authority (RDIA), Saudi Arabia, Riyadh, Reactivating & Rebuilding of Existing Labs Initiative number (13262-Tabuk-2023-UT-R-3-1-HW).

## Author contributions

**Conceptualization:** Gomathi Rajagopal, Balajee Ramachandran, Paradesi Deivanayagam.

**Data curation:** Balajee Ramachandran.

**Formal analysis:** Goyitom Gebremedhn Gebru, Gomathi Rajagopal, Balajee Ramachandran, Boomi Pandi, Saravanan Muthupandian.

**Investigation:** Paradesi Deivanayagam.

**Methodology:** Gomathi Rajagopal, Balajee Ramachandran, Venkatesan Srinivasadesikan, Boomi Pandi.

**Software:** Venkatesan Srinivasadesikan, Boomi Pandi, Rajamanikandan Sundarraj.

**Supervision:** Balajee Ramachandran, Paradesi Deivanayagam.

**Validation:** Balajee Ramachandran.

**Visualization:** Paradesi Deivanayagam.

**Writing – original draft:** Gomathi Rajagopal, Balajee Ramachandran, Venkatesan Srinivasadesikan, Saravanan Muthupandian.

**Writing – review & editing:** Goyitom Gebremedhn Gebru, Paradesi Deivanayagam.

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
