## [Decision Letter · Decision Letter 0]

16 Aug 2024

PONE-D-24-27790Induction of Zinc conjugated with Doxorubicin for the prevention of aggregating β-catenin in the Wnt signaling pathway investigated through Computational approaches.PLOS ONE

Dear Dr. Gebru,

Thank you for submitting your manuscript to PLOS ONE. After careful consideration, we feel that it has merit but does not fully meet PLOS ONE’s publication criteria as it currently stands. Therefore, we invite you to submit a revised version of the manuscript that addresses the points raised during the review process.

We look forward to receiving your revised manuscript.

Kind regards,

Ahmed A. Al-Karmalawy, PhD

Academic Editor

PLOS ONE

A clean copy of the edited manuscript (uploaded as the new *manuscript* file).

4. PLOS requires an ORCID iD for the corresponding author in Editorial Manager on papers submitted after December 6th, 2016. Please ensure that you have an ORCID iD and that it is validated in Editorial Manager. To do this, go to ‘Update my Information’ (in the upper left-hand corner of the main menu), and click on the Fetch/Validate link next to the ORCID field. This will take you to the ORCID site and allow you to create a new iD or authenticate a pre-existing iD in Editorial Manager. Please see the following video for instructions on linking an ORCID iD to your Editorial Manager account: https://www.youtube.com/watch?v=_xcclfuvtxQ".

Reviewers' comments:

Reviewer's Responses to Questions

**Comments to the Author**

1. Is the manuscript technically sound, and do the data support the conclusions?

Reviewer #1: Yes

Reviewer #2: Yes

2. Has the statistical analysis been performed appropriately and rigorously? 

Reviewer #1: Yes

Reviewer #2: N/A

3. Have the authors made all data underlying the findings in their manuscript fully available?

Reviewer #1: No

Reviewer #2: No

4. Is the manuscript presented in an intelligible fashion and written in standard English?

Reviewer #1: No

Reviewer #2: Yes

5. Review Comments to the Author

Reviewer #1: Recommendation: Publish after major revisions noted.

The manuscript by Gebru and co-workers explored the role of free Doxorubidicin and zinc conjugated Doxorubidicin against β-catenin in treatment of cervical cancer. The study has been conducted purely from the computational standpoint where the authors have used Molecular Docking, Molecular Dynamics and Density Functional Theory (DFT) to understand the stabilizing factors present in both cases. While the manuscript furnishes some insightful data, I found the manuscript to be poorly written, difficult to follow with many inconsistencies. The manuscript may be published after rewriting it in a more compact manner. The abstract is also filled with redundant information and may be shortened. There are certain additional comments which are listed below:

1. The authors have carried out the DFT studies in a number of functionals. It is quite impressive. However, I would like to see the numbers in B3lYP-D3 as I think dispersion correction will play a significant role in these systems.

2. Have the authors considered QM/MM studies? I am curious as to what happens when the structures are subjected to the entire protein environment.

Reviewer #2: The overall paper is a very interesting read and is very well put together. Here are a few comments and clarifications:

• First part of paragraph 1 lacks references (eg: line 3, line 4)

• Page 5 paragraph 2: Reference missing for FDA claim and missing references in a fewer line after that

• Page 6: ChemDraw and Schrodinger have not been referenced

• On what basis was the Zinc conjugated to the doxorubicin? Was it mostly chemical intuition or is there previous studies that were done for it?

• How were the missing residues added to the PDB?

• What are the forcefields used for MD simulations?

• Is 50 ns enough simulation time? Were there replicates done for the simulation?

• What were the basis of choosing the specific model chemistries?

• Page 8: Missing reference for STRING Database

• Page 13 and 14: Figure7 can be separated into 2 figs or can be made smaller to accommodate in one page

6. PLOS authors have the option to publish the peer review history of their article (what does this mean? ). If published, this will include your full peer review and any attached files.

**Do you want your identity to be public for this peer review?** For information about this choice, including consent withdrawal, please see our Privacy Policy .

Reviewer #1: No

Reviewer #2: No

---

## [Author Response · Author response to Decision Letter 1]

19 Oct 2024

Response to Reviewers

Reviewer #1:

Recommendation: Publish after major revisions noted.

The manuscript by Gebru and co-workers explored the role of free Doxorubicin and zinc conjugated Doxorubicin against β-catenin in the treatment of cervical cancer. The study has been conducted purely from the computational standpoint where the authors have used Molecular Docking, Molecular Dynamics and Density Functional Theory (DFT) to understand the stabilizing factors present in both cases. While the manuscript furnishes some insightful data, I found the manuscript to be poorly written, difficult to follow with many inconsistencies. The manuscript may be published after rewriting it in a more compact manner. The abstract is also filled with redundant information and may be shortened. There are certain additional comments which are listed below:

1. The authors have carried out the DFT studies in a number of functionals. It is quite impressive. However, I would like to see the numbers in B3lYP-D3 as I think dispersion correction will play a significant role in these systems.

Response: We have calculated the binding energy at B3LYP-D3 level of theory in the gas phase and the results were added in the manuscript. The results of binding energy at B3LYP-D3 optimization are close agreement with M06 level of theory (See Table 1). However, after the addition of metal atom the binding energy result is close in agreement with the BHandHLYP and ωB97XD level of theory. It shows that the dispersion corrected method calculates the binding energy of metal complexes reasonably good and given in Page Number: 15

2. Have the authors considered QM/MM studies? I am curious as to what happens when the structures are subjected to the entire protein environment.

Response: Thanks for your comments.

Due to the limitation of software availability, we will execute the QM/MM studies in the future. However, the structure in the protein environment is addressed by Molecular dynamics simulation and the results are shown in MD simulation

Reviewer #2: The overall paper is a very interesting read and is very well put together. Here are a few comments and clarifications:

1. First part of paragraph 1 lacks references (eg: line 3, line 4)

Response: Thanks for your comments. We have inserted the reference in the line 3 and 4 and shown in Page Number: 2

2. Page 5 paragraph 2: Reference missing for FDA claim and missing references in a fewer line after that

Response: Thanks for your comments. We have included the references in that section and given in the page number: 5.

3. Page 6: ChemDraw and Schrodinger have not been referenced

Response: Thanks for your comments. The appropriate reference has been cited for Chemdraw and Schrodinger, it was shown in the Page Number: 6.

4. On what basis was the Zinc conjugated to the doxorubicin? Was it mostly chemical intuition or is there previous studies that were done for it?

Response: Thanks for your comments.

Gewirtz D. A. (1999). A critical evaluation of the mechanisms of action proposed for the antitumor effects of the anthracycline antibiotics adriamycin and daunorubicin. Biochemical pharmacology, 57(7), 727–741. https://doi.org/10.1016/s0006-2952(98)00307-4 propsed mechanisms by which doxorubicin acts in the cancer cell (i) intercalation into DNA and disruption of topoisomerase-II-mediated DNA repair and (ii) generation of free radicals and their damage to cellular membranes, DNA and proteins.

Zinc is essential for microorganisms, plants, and animals. Deprivation of zinc arrests growth and development and produces system dysfunction in these organisms. Zinc provides the structural integrity of proteins and the regulation of gene expression. Hence, we have conjugated the Zinc with Doxorubicin to understand insights to analyse how the Zn-drug conjugates can effectively target malignant cells and induce apoptosis.

5. How were the missing residues added to the PDB?

Response: Thanks for your comments.

The missing residues were added when preparing the protein using Protein Preparation Wizard of Schrodinger and shown in Page Number: 6

6. What are the forcefields used for MD simulations?

Response: Thanks for your comments.

The Molecular Dynamics Simulation was performed Desmond package with OPLS2005 force field and shown in Page Number:7.

7. Is 50 ns enough simulation time? Were there replicates done for the simulation?

Response: Thanks for your comments.

The Molecular Dynamics Simulation was performed for the period of 100ns as triplicates. The average interaction and their contributions are given in Figure 6 and 7 and it was shown in Page Number: 11 to 13.

8. What were the basis of choosing the specific model chemistries?

Response: Thanks for your comments

The chosen model of drug loading chemistry effectively combines enhanced stability, high loading efficiency, and controlled release, while enabling targeted delivery and maintaining biocompatibility. This comprehensive approach is designed to improve the efficacy of cancer therapies, optimize patient outcomes, and address the challenges associated with conventional drug delivery systems.

9. Page 8: Missing reference for STRING Database

Response: Thanks for your comments.

The reference for the STRING database were added and shown in Page Number: 8

10. Page 13 and 14: Figure 7 can be separated into 2 figs or can be made smaller to accommodate in one page

Response: Thanks for your comments

The Figures are renumbered. Hence, the figures are updated as Figure 6 and Figure 7 and shown in Page Number: 12. The RMSD figures of the Molecular Dynamics simulation are moved to supplementary material and numbered as S1 and S2.

---

## [Decision Letter · Decision Letter 1]

6 Nov 2024

PONE-D-24-27790R1Induction of Zinc conjugated with Doxorubicin for the prevention of aggregating β-catenin in the Wnt signaling pathway investigated through Computational approaches.PLOS ONE

Dear Dr. Gebru,

Thank you for submitting your manuscript to PLOS ONE. After careful consideration, we feel that it has merit but does not fully meet PLOS ONE’s publication criteria as it currently stands. Therefore, we invite you to submit a revised version of the manuscript that addresses the points raised during the review process.

We look forward to receiving your revised manuscript.

Kind regards,

Ahmed A. Al-Karmalawy, PhD

Academic Editor

PLOS ONE

Reviewers' comments:

Reviewer's Responses to Questions

**Comments to the Author**

1. If the authors have adequately addressed your comments raised in a previous round of review and you feel that this manuscript is now acceptable for publication, you may indicate that here to bypass the “Comments to the Author” section, enter your conflict of interest statement in the “Confidential to Editor” section, and submit your "Accept" recommendation.

Reviewer #1: All comments have been addressed

Reviewer #3: (No Response)

2. Is the manuscript technically sound, and do the data support the conclusions?

Reviewer #1: Yes

Reviewer #3: (No Response)

3. Has the statistical analysis been performed appropriately and rigorously? 

Reviewer #1: Yes

Reviewer #3: (No Response)

4. Have the authors made all data underlying the findings in their manuscript fully available?

Reviewer #1: Yes

Reviewer #3: (No Response)

5. Is the manuscript presented in an intelligible fashion and written in standard English?

Reviewer #1: Yes

Reviewer #3: (No Response)

6. Review Comments to the Author

Reviewer #1: The authors have addressed all the points raised by reviewer properly. The manuscript may be considered for publication in its current form.

Reviewer #3: Authors of the presented manuscript evaluated the computational affinity Doxorubicin and Zinc conjugated with Doxorubicin against β-catenin biotarget for identifying promising anti-cancer agents. Study is relevant to its field. Following comments and suggestions should be addressed:

1. Within docking study, authors should elaborate more on the polar hydrogen bond interactions. Hydrogen binding should further be presented within hydrogen bond distances as well as hydrogen bond angles since hydrogen bond depends on both. Authors should mention the Hydrogen bond angles as well as their distances, since the strength of hydrogen bonding is based on both parameters in a way to ensure the adequacy of optimum hydrogen bonding.

2. Authors should adopt positive controls or reported reference compounds for the β-catenin target throughout the in silico studies in a way to validate and ensure the successful biological translation of the computational findings.

3. At molecular dynamics study, authors should explore the comparative mm-PBSA/GBSA free binding energies for each ligand-target complex to highlight the nature (electrostatic, van der Waal, and solvation) and magnitude of binding the thing that would guide further compound’s development and optimization.

4. Further, authors are advised to provide overlays for the initial, middle, and final frames (at 0 ns, 50 ns, and 100 ns, respectively) for each ligand-protein complex across the molecular dynamics simulations. This approach would provide great insights regarding the time-evolution orientation/conformation changes for both the protein and bounded ligands as well as the conserved and reformed ligand-amino acid bindings and close-range contacts.

7. PLOS authors have the option to publish the peer review history of their article (what does this mean? ). If published, this will include your full peer review and any attached files.

**Do you want your identity to be public for this peer review?** For information about this choice, including consent withdrawal, please see our Privacy Policy .

Reviewer #1: No

Reviewer #3: **Yes**

---

## [Author Response · Author response to Decision Letter 2]

12 Dec 2024

Reviewer #3: Authors of the presented manuscript evaluated the computational affinity Doxorubicin and Zinc conjugated with Doxorubicin against β-catenin biotarget for identifying promising anti-cancer agents. Study is relevant to its field. Following comments and suggestions should be addressed:

1. Within docking study, authors should elaborate more on the polar hydrogen bond interactions. Hydrogen binding should further be presented within hydrogen bond distances as well as hydrogen bond angles since hydrogen bond depends on both. Authors should mention the Hydrogen bond angles as well as their distances since the strength of hydrogen bonding is based on both parameters in a way to ensure the adequacy of optimum hydrogen bonding.

Response: Thanks for your comments.

We have displayed the hydrogen bond interactions and highlighted the distance of the complex and given in the page number: 9,10, Table -1 and shown in Fig4, 5.

2. Authors should adopt positive controls or reported reference compounds for the β-catenin target throughout the in silico studies in a way to validate and ensure the successful biological translation of the computational findings.

Response: Thanks for your comments.

Yang et al., 2021 has comprehensively listed the β-catenin inhibitors along with their experimental values in their review. We have started the screening procedure with the two β-catenin inhibitors such as (i) PKF118-310 (IC50: 0.8 µM) and (ii) ZTM000990 (IC50: 0.64 µM) as a reference. The PKF118-310 has shown the binding energy -4.13 kcal/mol and ZTM000990 has shown -8.50 kcal/mol (~0.642 μM) (Figure S1). Hence, ZTM000990 is taken as a reference compound for the present study and given in Page Number: 11.

3. At molecular dynamics study, authors should explore the comparative mm-PBSA/GBSA free binding energies for each ligand-target complex to highlight the nature (electrostatic, van der Waal, and solvation) and magnitude of binding the thing that would guide further compound’s development and optimization.

Response: Thanks for your comments

MM/GBSA analysis

The Binding Free Energy was computed every 20th frame of the protein-ligand complex, and the average value was reported. The ZTM000990 has shown an average value of -52.84 kcal/mol followed that the Doxorubicin and Zn-Doxorubicin resulted in -48.25 kcal/mol, -57.36 kcal/mol, respectively. The study suggests that Zn-Doxorubicin has shown better interaction with the Val349, Lys354, and Arg386; Phe21 of Tcf-4. In contrast, the Doxorubicin has shown interaction with charged residues and less interaction with Phe21 (hydrophobic residue). The dGcoulomb for ZTM00990, Zn-Doxorubicin and Doxorubicin has shown -39.26, -68.93 and -35.66 kcal/mol respectively (Table – 2). This study shows that interaction energy between the protein and ligand is found optimized in the presence of Zn2+ and due to the presence of hydrophobic residues in the binding site, hence, the complex was identified as stable. The lowest energy of ΔGbind values indicates the compactness of the ligands inside the binding pocket of β-catenin.

Solvation and desolvation effects are considered a most effective approach to evaluate the global and local conformation of protein in the solvent model[46]. The solvation-free energy values were calculated for every residue, and it was averaged. The averaged solvation energy for ZTM-000990 has shown -5.35 kcal/mol, and Doxorubicin and Zn-Doxorubicin complex has shown -3.38 and -6.55 kcal/mol respectively. The difference between the Zn-Doxorubicin and ZTM-000990 has shown ~1.0 kcal/mol order of magnitude whereas the Doxorubicin has shown ~ 3 kcal/mol. However, the Zn2+ has a role of altering the conformation as well as attracting possible contacts to maintain stability and shown in page number: 16

4. Further, authors are advised to provide overlays for the initial, middle, and final frames (at 0 ns, 50 ns, and 100 ns, respectively) for each ligand-protein complex across the molecular dynamics simulation. This approach would provide great insights regarding the time-evolution orientation/conformation changes for both the protein and bounded ligands as well as the conserved and reformed ligand-amino acid bindings and close-range contacts.

Response: Thanks for your comments

We have extracted the structures of every 20ns, it has been overlaid with the starting structure. The RMSD values were observed for the same shown in Figure S3 and S4. The conformational changes occurred ~ 1 Å from the starting to end frames. Further, the Tcf-4 has changed a bit during simulation shown in Page number: 20 (Figure 11) and shown in Supplementary Figure S4, S5.

---

## [Decision Letter · Decision Letter 2]

16 Dec 2024

Induction of Zinc conjugated with Doxorubicin for the prevention of aggregating β-catenin in the Wnt signaling pathway investigated through Computational approaches.

PONE-D-24-27790R2

Dear Dr. Gebru,

We’re pleased to inform you that your manuscript has been judged scientifically suitable for publication and will be formally accepted for publication once it meets all outstanding technical requirements.

Kind regards,

Ahmed A. Al-Karmalawy, PhD

Academic Editor

PLOS ONE

Reviewers' comments:

Reviewer's Responses to Questions

**Comments to the Author**

1. If the authors have adequately addressed your comments raised in a previous round of review and you feel that this manuscript is now acceptable for publication, you may indicate that here to bypass the “Comments to the Author” section, enter your conflict of interest statement in the “Confidential to Editor” section, and submit your "Accept" recommendation.

Reviewer #3: All comments have been addressed

2. Is the manuscript technically sound, and do the data support the conclusions?

Reviewer #3: (No Response)

3. Has the statistical analysis been performed appropriately and rigorously? 

Reviewer #3: (No Response)

4. Have the authors made all data underlying the findings in their manuscript fully available?

Reviewer #3: (No Response)

5. Is the manuscript presented in an intelligible fashion and written in standard English?

Reviewer #3: (No Response)

6. Review Comments to the Author

Reviewer #3: (No Response)

7. PLOS authors have the option to publish the peer review history of their article (what does this mean? ). If published, this will include your full peer review and any attached files.

**Do you want your identity to be public for this peer review?** For information about this choice, including consent withdrawal, please see our Privacy Policy .

Reviewer #3: **Yes: **

---

## [Editor Report · Acceptance letter]

PONE-D-24-27790R2

PLOS ONE

Dear Dr. Gebru,

I'm pleased to inform you that your manuscript has been deemed suitable for publication in PLOS ONE. Congratulations! Your manuscript is now being handed over to our production team.

Kind regards,

on behalf of

Associate Professor Ahmed A. Al-Karmalawy

Academic Editor

PLOS ONE